# Effects of Fermented Tea Residue on Fattening Performance, Meat Quality, Digestive Performance, Serum Antioxidant Capacity, and Intestinal Morphology in Fatteners

**DOI:** 10.3390/ani10020185

**Published:** 2020-01-22

**Authors:** Xiaoqing Ding, Huaiyu Li, Zhiwei Wen, Yong Hou, Genliang Wang, Jinghui Fan, Lichun Qian

**Affiliations:** 1Key Laboratory of Animal Nutrition and Feed Science in East China, Ministry of Agriculture, College of Animal Sciences, Zhejiang University, Hangzhou 310058, China; 21817065@zju.edu.cn (X.D.); 21817082@zju.edu.cn (H.L.); 3190100233@zju.edu.cn (Z.W.); hou123650@163.com (Y.H.); 2Songyang Green Valley Tea Aroma Agriculture Co., Ltd., Hangzhou 323000, China; 15551835003@163.com; 3Hangzhou Academy of Agricultural Sciences, Hangzhou 310021, China; zigugu@hz.cn

**Keywords:** fermented tea residue, fatteners, fattening performance, digestion performance, meat quality, serum antioxidant capacity, intestinal morphology

## Abstract

**Simple Summary:**

Tea residue is the remaining residue of tea after drinking or deep processing. With the annual consumption of tea, a large amount of discarded tea residue is produced. If the tea residue is not effectively utilized, it will cause not only environmental pollution, but also represent a waste of resources. In this study, strains of *Bacillus subtilis*, *Aspergillus niger*, and *Saccharomyces cerevisiae* were used to produce fermented tea residue (FTR). Different levels of FTR were used instead of corn and soybean meal. The resulting product was used to feed pigs bred for rapid growth, also known as fatteners. We explored its effects on growth performance, digestion performance, meat quality, serum antioxidant capacity, and intestinal morphology in fatteners. The aim is to provide a scientific basis for the future use of FTR as a feed material.

**Abstract:**

This study investigated the dietary supplementation of tea residue fermented by *Bacillus subtilis*, *Aspergillus niger*, and *Saccharomyces cerevisiae*, to explore its effects on growth performance, digestion performance, meat quality, serum antioxidant capacity, and intestinal morphology in pigs bred for rapid growth, also known as fatteners. One hundred and ninety-two healthy “Duroc × Landrace × Yorkshire” ternary hybrid pigs (body weight 70 ± 1.0 kg) were randomly divided into four groups according to the feeding test requirements, with four replicates in each group, and 12 fatteners per replicate. The control group (CG) was fed the basal diet. Treatments 1 (T1), 2 (T2), and 3 (T3), comprising ratios of 10%, 15%, and 20% of tea residue were added to the basal diet. The test period was 60 days. The results showed that supplementation of FTR in fatteners’ diets increased final body weight (FBW), average daily gain (ADG), and feed conversion ratio (FCR) in the T1 and T2 groups (*p* < 0.05). Compared with the other groups, the lightness (L*) and pH were significantly affected in the T2 group (*p* < 0.05). Compared with the CG, dietary supplementation of FTR significantly increased the nutrient digestibility of crude protein (CP), ether extract (EE), calcium (Ca), and phosphorus (P), improved the lipase and trypsin activities, and reduced drip loss and the shear force of fatteners (*p* < 0.05). Glutathione peroxidase (GSH-Px) and total antioxidant capacity (T-AOC) were significantly increased in the T2 and T3 groups compared with the other groups (*p* < 0.05). Supplementation of FTR in the jejunum significantly increased the villi height of the T2 group and the ratio of villi height to crypt depth of the FTR groups. Compared with the other two groups, the T2 and T3 groups significantly reduced the ratio of the villous height to crypt depth in the duodenum (*p* < 0.05). In conclusion, the tea residue after fermentation was shown to have beneficial effects on the fattening performance, digestion performance, meat quality, serum antioxidant capacity, and intestinal morphology of fatteners.

## 1. Introduction

Tea is one of the most popular and inexpensive beverages in the world. According to FAO statistics, world tea production increased to about 568 million tons in 2018 [1]. China is the world’s major producer and exporter of tea [2]. Tea residue is the remaining residue of tea after drinking or deep processing. However, with the continuous increase in the annual consumption of tea and the growth of the deep processing technology in the tea industry, a large amount of discarded tea residue is produced. Currently, tea residue is mainly produced by the industrial extraction of functional ingredients (e.g., tea polyphenols) and residues from beverage manufacturing [3]. A lot of tea residue is discarded and wasted [4]. If tea residue is discarded or incinerated, it not only pollutes the environment, but is also a waste of resources. The resource utilization of tea residue has become a problem to be solved [3].

The large-scale development of the pig industry requires large amounts of protein feed. However, due to the high prices of fish and soybean meal, the cost of protein feed has gradually increased, which has had a considerable impact on the pig industry. Tea residue fermented by *Bacillus subtilis*, *Aspergillus niger*, and *Saccharomyces cerevisiae* is not only rich in protein, but also contains amino acids, tea polyphenols, crude fiber, and trace elements that pigs require. It has been reported that tea residue has the potential to be used as protein feed because of its higher crude protein (21–28%) content [5]. Newey and Smyth [6] used alkali-soluble acid precipitation to extract protein from tea residue as a feed ingredient, thereby improving the enzyme activity and pancreatic function of the gastrointestinal tract of fatteners. In addition, lipid peroxidation could produce toxic compounds such as peroxides and aldehydes, leading to meat quality degradation [7], and the tea catechins and tea polyphenols in FTR have strong antioxidant activities. Adding FTR to pig diets can effectively improve the antioxidant status and prevent lipid oxidation in pork [8]. Tea polyphenols can better enhance the fattening performance of pigs, and can promote central nervous excitement, which is beneficial to the metabolism of pigs, significantly increasing their daily weight gain and shortening the slaughter time of fatteners [9]. The fiber content in feed for ruminants is generally around 17%; therefore, FTR may also become a ruminant feed ingredient, and will be widely used in the future.

At present, many reports have proposed tea residue as a feed ingredient directly to the animal diet, or a small amount of it as a functional additive. However, there are few reports in which the tea residue is first fermented and then used as the main ingredient of the diets of livestock and poultry. Therefore, the main aim of this study was to use tea residue as the primary fermentation substrate. The fermentation of tea residue was performed by mixing strains of *Bacillus subtilis*, *Aspergillus niger*, and *Saccharomyces cerevisiae*. The FTR was added to the diet of pigs to explore its effects on fattening performance, digestion performance, meat quality, serum antioxidant capacity, and intestinal morphology.

## 2. Materials and Methods

All procedures were carried out based on protocols approved by the Animal Care Advisory Committee of Zhejiang University.

### 2.1. Fermented Tea Residue Sample

Tea residue was provided by Songyang County Green Valley Tea Town Agricultural Technology Co., Ltd. (Hangzhou, China). *Aspergillus niger*, *Saccharomyces cerevisiae*, and *Bacillus subtilis* were provided by the Institute of Feed Science of Zhejiang University (Hangzhou, China). The tea residue was used as a fermentation substrate, and the three strains of *Aspergillus niger*, *Saccharomyces cerevisiae*, and *Bacillus subtilis* bacteria were uniformly added to the tea residue in a ratio of 1:1:2 (addition amount of 5 × 10^7^ cfu/g), respectively. Fermentation at 30 °C and 55% water content for 72 h was applied to obtain the FTR required for our test. The feed was processed into pellets by Songyang County Green Valley Tea Town Agricultural Technology Co., Ltd. (Hangzhou, China). After the tea residue was fermented by a microbial strain, its nutritional indicators were determined using the methods recommended by AOAC [10]. The nutritional indicators were 41.20% crude protein, 4.71% crude fat, 11.08% crude fiber, 11.04% crude ash, 4.37% calcium, and 0.56% total phosphorus. It also contained chemical ingredients such as polyphenols, catechins, proteins, amino acids, tea saponins, vitamins etc.

### 2.2. Animals and Diets

In this study, 192 fatteners (Duroc × Landrace × Yorkshire, Duroc is the terminal boar, Landrace is the male parent, and Yorkshire is the female parent. Duroc is from the United States, Landrace is from Denmark, and Yorkshire is from North England.) with an average body weight of 70 ± 1.0 kg were randomly assigned to four dietary treatments with four replicates of 12 pigs each. The experimental diet was formulated based on the nutrient requirements established by the National Research Council [11]. The four treatments were CG (basal diet), T1 (basic diet + 10% FTR instead of basic diet feed ingredients 5% soybean meal and 5% corn), T2 (basic diet + 15% FTR instead of basic diet feed ingredients 7.5% soybean meal and 7.5% corn), and T3 (basic diet + 20% FTR instead of basic diet feed ingredients 10% soybean meal and 10% corn). The basic diet composition and nutritional level of the test are shown in Table 1. The pretest period was seven days and the total test period was 60 days. Feed was given twice a day at 7:00 am and 3:00 pm. The feed intake and fattening performance of fatteners were recorded daily during the test, and the number of deaths was recorded in detail. All fatteners were weighed at the beginning and the end of the experiment. The average daily body weight gain (ADG), average daily feed intake (ADFI), and feed conversion ratio (FCR) were calculated at different stages.

### 2.3. Sample Collection

At the end of the feeding experiment, the fatteners were fasted for 24 h, after which blood samples were collected from the anterior jugular vein using a vacuum tube. The blood samples were centrifuged at 3500× *g* for 15 min at 4 °C, and then serum was separated and stored at −20 °C for further analysis. Then fatteners were slaughtered according to the recommendations of the animal welfare literature [12]. The *longissimus dorsi* muscle was taken from 12th–13th intervertebral in order to measure the lightness (L*), redness (a*), yellowness (b*), and pH, before being wrapped in plastic wrap, put it into a sample bag, and then put into an icebox for further testing. Samples of about 2 cm in length were taken from the duodenum, the middle end of the jejunum, and the rear end of the ileum of the fatteners. The contents of the duodenum, jejunum, and ileum were loaded into a 1.5 mL eppendorf tube. The sample bag and the eppendorf tube were wrapped with a suitable tin foil and placed into a liquid nitrogen tank for temporary storage. The samples later stored in a laboratory −80 °C refrigerator.

### 2.4. Meat Quality

About 45 min and 24 h after slaughter, a longer cross-section was removed (from the 12th to the 13th rib). Meat color (L*, a*, b*) was measured with a Minolta chromameter (CR300, Minolta Camera CO., Osaka, Japan). Muscle pH was measured by using a pH meter. The pH of the upper, middle, and back parts of the *longissimus dorsi* was measured separately, and the average value was taken as the pH of the sample. Within 2 h after slaughter, the *longissimus dorsi* were taken from two fatteners, and the dripping loss was determined by the suspension method [13]. The cooking loss of pork was determined as follows: 120 g (W1) of the *longissimus dorsi* was weighed with a balance. The meat sample was heated in a water bath at 100 °C for 40 min, then the surface was removed with a pair of tweezers and allowed to dry, cooled to room temperature, and weighed again (W2). Cooking loss (%) = 100 × (W1 − W2)/W1. After the pig was slaughtered, the pork sample was thawed at 4 °C for 24 h, placed in a sealed bag, and then heated in a water bath at 80 °C for about 30–40 min. Meat samples were taken out when the temperature reached 70 °C. The meat sample was cooled for 30 min. The cooked pork was cut into 1 × 1 × 3 cm^2^ parallel to the direction of the muscle fibers using a Tensipresser (TTP-50BXII, Taketomo Electric Corp., Tokyo, Japan) to measure tenderness.

### 2.5. Digestibility Trial and Digestive Enzyme

In the last week of the feeding experiment, the indirect method was used for the digestion test, i.e., 0.5% Cr_2_O_3_ was added to the fatteners’ diet. Feces were collected every day at 4 pm over three days. Nitrogen was fixed according to 10 mL of 10% H_2_SO_4_ per 100 g of feces. Then, the feces were placed into an oven at 65 °C for drying and regaining. Upon reaching numerical stability, the feces were weighed, and after being crushed through a 40-mesh screen, samples were obtained. The feed was also treated in the same way. The feces and diets were analyzed for dry matter (DM, method 930.15) and crude protein (Cp, 6.25 × N; method 984.13). The contents of ash (method 942.05), ether extract (EE, method 920.39), phosphorus (P, method 965.17), and calcium (Ca, method 968.08) were conducted according to the methods of the AOAC [10]. The activities of amylase, lipase, and trypsin in the digesta were analyzed using commercially-available amylase, lipase, and trypsin assay kits, according to the manufacturer’s instructions (ANG-SH-21041 and ANG-SH-21052, Nanjing Angle Gene Biotechnology, Nanjing, China).

### 2.6. Determination of Antioxidant Capacity Indicators

Serum superoxide dismutase (SOD) activity was determined by the pyrogallol auto-oxidation method. Catalase (CAT) activity was determined by the molybdate method, and serum glutathione peroxidase (GSH-Px) activity was determined by colorimetry. The serum total antioxidant capacity (T-AOC) was determined by Fe^3+^ reduction. The content of malondialdehyde (MDA) was measured with the thiobarbituric acid method using corresponding assay kits (Jiancheng Bioengineering Institute of Biological Engineering, Nanjing, China) with UV-VIS Spectrophotometer (UV1100, MAPADA, Shanghai, China), according to the manufacturer’s instructions.

### 2.7. Intestinal Morphology

An intestinal morphology analysis was carried out according to the method described by Choe et al. [14]. The duodenum, jejunum, and ileum samples were fixed for 24 h and dehydrated, and then embedded in paraffin. The embedded intestinal tissue pieces were cut into 5-μm-thick sections with hematoxylin-eosin (H&E). Staining was performed, and the histological morphologies were observed under a light microscope. Differences in intestinal morphology were compared using an Olympus BX51 microscope (4 × 1) (Olympus, Tokyo, Japan). Villi height (VH) was measured from the tip of the villi to the villus crypt junction, and crypt depth (CD) was defined as the depth of the invagination between two villi. Both the VH and the CD were determined using the Image-Pro Plus software (Media Cybernetics, Silver Spring, MD, USA) as described by Touchette et al. [15]. The ratio of villus height to crypt depth were then calculated.

### 2.8. Statistical Analysis

All data were analyzed by one-way analysis of variance (ANOVA) using SPSS statistical software (Ver. 20.0 for Windows, SPSS, Inc., Chicago, IL, USA). Differences among treatments were examined using the Tukey-Kramer’s multiple range tests, which were considered significant when the *p*-value was less than 0.05. The results are presented as means alongside their pooled standard errors of means (SEM).

## 3. Results

### 3.1. Fattening Performance

The results of the fattening performance of pigs are presented in Table 2. There was no difference in ADFI among all groups (*p* > 0.05). Compared with the CG and T3 group, the ADG and FBW of the T1 and T2 groups improved significantly (*p* < 0.05). The fatteners fed FTR diet had significantly lower FCR than those fed the basic diet.

### 3.2. Meat Quality

The results of meat quality are shown in Table 3. There were no differences in the a*, b*, cooking loss among all groups (*p* > 0.05). The L* of the T2 group was lower than that of the other three groups (*p* < 0.05). Compared with the CG group, the FTR groups significantly reduced drip loss and shear force, and increased pH at 45 min (*p* < 0.05).

### 3.3. Digestive Performance

The effect of the nutrient digestibility of fatteners is shown in Table 4. Compared with the CG group, the treatment groups with FTR significantly increased the digestibility of crude protein (CP), ether extract (EE), calcium (Ca), and phosphorus (P) (*p* < 0.05). No differences were observed in dry matter (DM) and organic matter (OM) among all groups (*p* > 0.05).

The effect of intestinal digestive enzyme activity in fatteners is shown in Table 5. Compared to the CG group, dietary supplement FTR significantly increased trypsin (*p* < 0.05), while there was no difference in amylase among the groups (*p* > 0.05). The lipase of the T2 and T3 groups were significantly higher than those in the other two groups (*p* < 0.05).

### 3.4. Antioxidant Capacity

The effects of the serum antioxidant capacity are presented in Table 6. The activity of glutathione peroxidase (GSH-Px) and total antioxidant capacity (T-AOC) of the T2 and T3 groups were significantly higher than those of the other two groups (*p* < 0.05). Compared with the CG group, dietary supplement FTR decreased MDA, and increased superoxide dismutase (SOD) and catalase (CAT), but the differences were not statistically significant (*p* > 0.05).

### 3.5. Intestinal Morphology

The results of the intestine morphology of fatteners are presented in Table 7. Compared with the CG and T1 groups, the ratio of the villous height to crypt depth of duodenum in the T2 and T3 groups increased significantly (*p* < 0.05); however, the VH and CD were not significantly affected in the duodenum (*p* > 0.05). The VH of the T2 group was significantly higher than that of the other three groups (*p* < 0.05). Compared with the CG group, dietary supplement FTR significantly increased the ratio of villi height to crypt depth of the jejunum. Supplementation of FTR decreased the CD in the jejunum and increased the VH and the ratio of the villous height to crypt depth in the ileum (*p* > 0.05).

## 4. Discussion

### 4.1. Fattening Performance

Tea residue fermented using *Bacillus subtilis*, *Aspergillus niger*, and *Saccharomyces cerevisiae* could reduce the content of antinutrient factors, as well as their adverse effects on fatteners [16]. In addition, the nutrients contained in FTR can be initially decomposed, reducing the wear and tear of the intestines, which is conducive to the absorption of nutrients in the intestines of pigs. The results of the current study showed that the supplementation of FTR significantly increased ADG, FBW, and feed efficiency in fatteners. This is in accordance with the reports of Yang et al. [17] that tea byproducts are a rich source of crude protein, which may have a beneficial effect on animal feed intake and weight gain. In addition, Ahmed et al. [18] suggested that the feed conversion ratio was improved by adding green tea byproducts. The growth-promoting effect of FTR may be related to the antioxidant effect of tea polyphenols and catechins [8]. Tea polyphenols and catechins could alleviate the oxidative damage of active oxygen to body lipids, improve the fat utilization of fatteners, and thereby promote the growth of fatteners. Moreover, the tea saponin in FTR could also increase the permeability of the intestinal mucosa and promote the absorption of nutrients, thereby promoting the growth of fatteners and improving feed efficiency. Conversely, Ko et al. [19] showed that dietary supplementation with green tea probiotics has no effect on the growth performance of finishing pigs. These differences may be due to the differences in tea species, fermentation procedures, and concentrations of tea.

### 4.2. Meat Quality

Color and the capacity of the meat to hold or bind water are highly variable and vital attributes of pork quality; pH is an important factor affecting them. When the final pH of pork is low, it results in pale color and reduced water retention capacity [20,21]. In the present study, dietary FTR supplementation had no significant influence on some parameters related to meat quality, such as a*, b*, and cooking loss. The pH of the FTR groups gradually increased and was higher than that of the CG group. The results of this experiment indicated that FTR has a protective effect on pork antioxidation and the prevention of carcass tissue cell membrane damage, and can inhibit muscle glycogenase and rancidity [22]. This has important implications for extending the shelf life of meat. Joo et al. [23] pointed out that pork with too high or low L* values post mortem is considered abnormal; for example, meats with a L* lower than 43 are considered DFD (dark, firm, and dry), but meats with L* higher than 50 are considered PSE (pale, soft, and exudative). In this study, the evaluated meat quality in the FTR groups was at normal levels. The experimental group significantly reduced drip loss and shearing force, i.e., the FTR increased the moisture content in pork and kept it tender, which is in line with research conducted by Jun et al. [24]. At the same time, the color of the meat was improved, which enhanced the water-tightness, changed the flavor and tenderness, and made it juicy [25,26,27]. Pig diet supplement FTR can make pork contain more umami and significantly improve the taste of pork.

### 4.3. Digestive Performance

Previous studies have shown that increasing the digestibility of nutrients can promote livestock digestion and improve their growth. The apparent digestibility of dietary nutrients in pigs can reflect their ability to digest and absorb feed [28]. In this study, the nutrient digestibility of CP, EE, Ca, and P in fatteners of each group supplemented with FTR was significantly improved compared with the CG group. This is similar to the results previously reported by Feng et al. [29], i.e., that supplementation of fermented soybean meal had beneficial effects on the apparent digestibility of finishing pigs. In the current study, the FTR in this test was fermented through *Bacillus subtilis*, *Aspergillus niger*, and *Saccharomyces cerevisiae*. These strains are beneficial to the digestion and absorption of intestinal nutrients in animals [30]. Hua et al. [31] found that the compound gastrointestinal regulator of *Bacillus subtilis* can improve the apparent digestibility of CP, EE, and total P in piglets, which was significantly different from the CG group. Digestive enzymes are special proteins that promote the degradation of food in the digestive tract, including proteases, lipases, and amylases. The level of digestive enzyme activity in an animal’s intestine reflects the strength of its digestive ability and determines the efficiency of feed utilization [32]. In tests using *Bacillus subtilis* Z-27 instead of antibiotics, piglet intestinal protease, amylases, and lipase activity were significantly increased [33]. This is in accordance with significantly increased lipase and trypsin activity in the treatment groups with FTR added to the diet in the presented experiment. Previous studies have demonstrated that *Bacillus subtilis* itself can secrete active digestive enzymes such as proteases, lipases, and amylases, which may lead to increased digestive enzyme activity with the feed entering the gastrointestinal tract [34,35].

### 4.4. Antioxidant Capacity

The rate and extent of the oxidation of fatteners depend on the antioxidant capacity. T-AOC, SOD, CAT, GSH-Px, and MDA all undergo free radical metabolism in the body, which directly or indirectly reflects the functional status of the antioxidant system. The antioxidant index can reflect the body’s scavenging of free radicals and the damage caused thereby to the body [36,37]. MDA is a product of lipid oxidation, which can reflect the degree of lipid peroxidation in the body and indirectly reflect the degree of cell damage [38]. In this study, the supplementation of FTR reduced the activities of MDA. This was consistent with previous observations, i.e., that green tea catechin decreased the MDA content in the serum of mice [39]. Freese et al. [40] reported that green tea reduced plasma MDA in humans. T-AOC is a comprehensive evaluation index of the body’s antioxidant system. SOD can catalyze the conversion of high-activity superoxide anion into low-activity H_2_O_2_, reducing the damage of superoxide anions to the body. The main role of GSH-Px is to remove lipid peroxides, and it can replace CAT to remove H_2_O_2_ when CAT content is very low. CAT participates in the process of reactive oxygen metabolism in animal bodies [37,41]. In the present study, the serum GSH-Px and T-AOC of fatteners fed with FTR increased significantly in the T2 and T3 groups, while the content of MDA decreased, and the activity of CAT and SOD increased. These findings are similar to those of Li et al. [42], who reported that tomato residue-fermented feed significantly increased the activity of SOD and CAT in Xinjiang brown cattle serum, increased T-AOC activity, and reduced MDA content. The results of this test indicate that adding fermented tea residue to the diet could improve the antioxidant performance of fatteners’ serum.

### 4.5. Intestinal Morphology

The intestine is not only the largest place for digestion and absorption into the body, but also the largest immune organ in the body. It constitutes the first barrier to prevent intestinal pathogens from invading [43]. The VH and CD are specific reflections of intestinal function. The VH of the intestine can reflect the digestive function of the nutrients therein. The higher the VH of intestinal, the larger the intestinal absorption area, the more mature the cells, and the stronger the nutrient absorption capacity [44,45]. The CD can reflect the rate of cell production. The crypts become shallower, indicating an increase in the rate of mature cells and enhanced secretory function [46]. The ratio of villus height to crypt depth can comprehensively reflect the function of the small intestine. Decreased ratios indicate damaged mucosa and reduced digestion and absorption capacity, and are often accompanied by diarrhea and impeded growth [47]. In the present study, the addition of FTR to the diet increased the VH and ratio of villi height to the crypt depth of the duodenum, jejunum, and ileumm and reduced the CD of the duodenum and jejunum, which was basically consistent with the results of previous studies. Missotten et al. [48] found that fermented feed could significantly increase the VH of the intestine, increase the surface area of the small intestine, and enhance the digestion and absorption of nutrients. Similar findings have been reported in the studies of Choi et al. [49], who demonstrated that adding fermented products to a piglet diet could increase the VH of duodenum, jejunum, and ileum, but that it did not significantly affect the CD. The results of this experiment showed that FTR could improve the intestinal environment, enhance the digestion and absorption of fatteners, and thereby increase the apparent digestibility of feed while maintaining the integrity of pig intestinal mucosa and improving their immunity.

## 5. Conclusions

In conclusion, dietary supplementation with FTR improved fattening performance, digestion performance, and the intestinal morphology of fatteners by increasing nutrient digestibility and digestive enzyme activity, while improving pork quality by enhancing the antioxidant performance of fatteners. Although this study demonstrated the positive effects of FTR on fatteners, further research is needed to establish the broad application of such additives, including optimal dose levels in feed, in order to obtain the maximum effect.

## Figures and Tables

**Table 1 animals-10-00185-t001:** Composition and nutrition value of experimental feed (air-dry basis, %).

Items	Experimental Groups			
	CG	T1	T2	T3
Ingredient				
Corn	70	65	62.5	60
Soybean meal	25	20	17.5	15
Fermented tea residue	0	10	15	20
Soybean oil	1	1	1	1
Dicalcium phosphate	0.2	0.2	0.2	0.2
Limestone	1.5	1.5	1.5	1.5
Salt	0.3	0.3	0.3	0.3
Premix **	2	2	2	2
Calculated Nutritional Value				
Digestive energy (MJ/kg)	14.20	13.61	13.31	13.02
Crude protein	17.24	18.63	19.33	20.03
Calcium	0.56	0.55	0.54	0.53
Available phosphorus	0.49	0.47	0.46	0.45
Lysine	0.95	0.99	1.03	1.05

** Premix: 1 kg of premix contained 16,000 IU VA, 3000 IU VD3, 100 IU VE, 2 mg VK3, 2 mg VB1, 9 mg VB2, 1.5 mg VB6, 0.02 mg VB12, 15 mg pantothenic acid, 40 mg nicotinic acid, 0.3 mg folic acid, 0.08 mg VB7, 600 mg of choline, 60 mg Zn, 70 mg Fe, 10 mg Cu, 20 mg Mn, 0.3 mg I, 0.3 mg Se. The feed did not contain any antibiotics.

**Table 2 animals-10-00185-t002:** Effect of fermented tea residue supplementation on fattening performance of fatteners.

Items	Experimental Groups				SEM	*p*-Value
	CG	T1	T2	T3		
**IBM (Kg)**	**69.85**	70.72	70.70	70.43	0.32	0.068
FBW (Kg)	122.14 ^b^	124.1 ^a^	125.87 ^a^	122.8 ^b^	1.32	0.015
ADG (g/d)	861.86 ^b^	902.05 ^a^	904.5 ^a^	868.92 ^b^	7.21	0.045
ADFI (Kg/d)	2.99	2.87	2.84	2.82	0.031	0.205
FCR	3.47 ^a^	3.18 ^bc^	3.14 ^c^	3.25 ^b^	0.043	0.032

^a–c^ Means with different superscripts in the same row indicate significant difference (*p* < 0.05).

**Table 3 animals-10-00185-t003:** Effect of fermented tea residue supplementation on meat quality of fatteners.

Items	Experimental Groups				SEM	*p*-Value
	CG	T1	T2	T3		
pH_45min_	6.11 ^a^	6.29 ^b^	6.45 ^b^	6.34 ^b^	0.38	0.030
pH_24h_	5.69 ^a^	5.77 ^ab^	5.97 ^b^	5.81 ^ab^	0.15	0.045
L*_45min_	51.39^a^	47.59 ^ab^	46.37 ^b^	49.10 ^a^	0.61	0.014
L*_24h_	51.67 ^a^	48.85 ^b^	47.59 ^b^	49.99 ^a^	0.59	0.048
a*_45min_	8.78	9.21	9.42	9.65	0.28	0.087
a*_24h_	7.64	8.19	8.36	8.48	0.44	0.079
b*_45min_	7.35	7.21	6.51	6.25	0.18	0.830
b*_24h_	10.44	10.21	8.97	9.16	0.44	0.411
Drip loss (%)	4.06 ^a^	3.54 ^b^	3.65 ^b^	3.07 ^c^	0.10	0.026
Cooking loss (%)	46.52	45.82	43.90	44.75	2.80	0.640
Shear force (kg)	5.97 ^a^	3.90 ^b^	4.12 ^b^	4.70 ^b^	0.96	0.007

^a–c^ Means with different superscripts in the same row indicate significant difference (*p* < 0.05).

**Table 4 animals-10-00185-t004:** Effect of fermented tea residue supplementation on nutrient digestibility of fatteners (%).

Items	Experimental Groups				SEM	*p*-Value
	CG	T1	T2	T3		
DM	84.56	85.69	83.78	84.35	0.69	0.590
CP	80.15 ^b^	82.42 ^a^	83.04 ^a^	82.35 ^a^	0.26	0.042
EE	33.95 ^b^	50.36 ^a^	47.76 ^a^	50.56 ^a^	1.82	<0.001
OM	83.91	83.56	84.68	82.95	0.77	0.141
Ca	32.39 ^c^	41.53 ^b^	55.89 ^a^	38.91 ^b^	1.58	<0.001
P	37.85 ^b^	48.83 ^a^	48.81 ^a^	48.72 ^a^	2.82	0.005

^a–c^ Means with different superscripts in the same row indicate significant difference (*p* < 0.05).

**Table 5 animals-10-00185-t005:** Effect of fermented tea residue supplementation on intestinal digestive enzyme activity of fatteners.

Items	Experimental Groups				SEM	*p*-Value
	CG	T1	T2	T3		
Amylase, U/mgprot	28.46	26.73	27.79	28.27	0.61	0.436
Lipase, U/mgprot	42.56 ^b^	43.99 ^b^	68.47 ^a^	69.74 ^a^	0.97	0.029
Trypsin, U/mgprot	60.45 ^c^	88.47 ^b^	109.10 ^a^	89.33 ^b^	2.60	0.016

^a–c^ Means with different superscripts in the same row indicate significant difference (*p* < 0.05).

**Table 6 animals-10-00185-t006:** Effect of fermented tea residue supplementation on serum antioxidant capacity of fatteners.

Items	Experimental Groups				SEM	*p*-Value
	CG	T1	T2	T3		
CAT (U/mL)	2.82	4.12	4.13	3.07	0.61	0.360
SOD (U/mL)	186.80	213.45	213.15	213.75	5.88	0.459
MDA (nmol/mL)	8.78	8.39	8.06	7.39	0.95	0.617
GSH-Px (U/mL)	118.62 ^b^	129.30 ^ab^	143.49 ^a^	139.74 ^a^	3.16	0.006
T-AOC (U/mL)	2.21 ^b^	2.35 ^b^	5.90 ^a^	5.24 ^a^	0.39	0.001

^a–c^ Means with different superscripts in the same row indicate significant difference (*p* < 0.05).

**Table 7 animals-10-00185-t007:** Effect of fermented tea residue supplementation on intestinal morphology of fatteners.

Items	Experimental Groups				SEM	*p*-Value
	CG	T1	T2	T3		
Duodenum						
VH (mm)	0.28	0.29	0.33	0.32	0.011	0.205
CD (mm)	0.17	0.16	0.14	0.15	0.009	0.621
VH: CD	1.65^c^	1.81 ^bc^	2.34 ^a^	2.13 ^b^	0.079	0.027
Jejunum						
VH (mm)	0.29 ^b^	0.31 ^b^	0.37^a^	0.33 ^ab^	0.013	0.040
CD (mm)	0.16	0.14	0.15	0.15	0.005	0.784
VH: CD	1.81^c^	2.21^b^	2.47^a^	2.20^b^	0.080	0.012
Ileum						
VH (mm)	0.29	0.32	0.33	0.34	0.013	0.476
CD (mm)	0.15	0.16	0.16	0.17	0.005	0.214
VH: CD	1.93	2.00	2.06	2.00	0.072	0.887

^a–c^ Means with different superscripts in the same row indicate significant difference (*p* < 0.05).

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
