# Peer review of "Effects of Fermented Tea Residue on Fattening Performance, Meat Quality, Digestive Performance, Serum Antioxidant Capacity, and Intestinal Morphology in Fatteners"

_animals, 2020, doi:10.3390/ani10020185_

Round 1
Reviewer 1 Report
English redaction should be improved (eg. Both present and past tenses verbs are used)
In Materials and Methods:
-It is necessary to show chemical composition of the fermented tea residue used and support discussion with that information
-There are some missing number of samples specified
-Declare why and what for Cr2O3 was included in diets
In Discussion some paragraphs alude other by-products, however did not explain how chemical components included also in Fermented tea residue do work. Avoid speculations.

Author Response
January 11, 2020
To the Editorial Office of the journal Animals
Dear Professors :
Thank you the reviewers for their constructive comments concerning our manuscript “Effects of fermented tea residue on fattening performance, meat quality, digestive performance, serum antioxidant capacity, and intestinal morphology in fattening pigs” (ID: animals-672334). Those comments are very valuable and helpful for revising our paper and guiding our researches. We have studied those comments carefully and have made corrections which we hope meet with approval. Revised portions are marked colored in the paper. The following is a point-to-point response to the editors' comments and recommendations. The main corrections in the paper and the response to the reviewer’s comments are as following:
Response to Reviewer 1 Comments and Reviewer 1’s comments were highlighted by using the Red Colored.
Point 1: English redaction should be improved (eg. Both present and past tenses verbs are used).
Response 1: Thank you for your comment. We have carefully revised the grammar and some detail errors in our manuscript and marked it with different colors according to all the referee suggestions.
Point 2: It is necessary to show chemical composition of the fermented tea residue used and support discussion with that information
Response 2: We have added the main ingredients of fermented tea residue in the manuscript (Line 103-107). Our previous research has shown that the crude protein, reducing sugar, protease of tea residue is significantly improved after fermentation by Bacillus subtilis, Aspergillus niger and Saccharomyces cerevisiae. It also contains tea polyphenols, plant alkaloids, and protein., amino acids, vitamins and other chemical ingredients needed by animals. (Ding et al.,2019. Pending papers). In addition, we made additions and appropriate changes during the discussion (Line 283-291).
Point 3: There are some missing number of samples specified.
Response 3: Sample number means sample marking. We have all supplemented the missing number of samples specified and modified the unreasonable expression according to your suggestion (Line 136-137, Line 139-141). Any change to our manuscript following your suggestion was highlighted by using the Red Colored Text.
Point 4: Declare why and what for Cr2O3 was included in diets
Response 4: Pigs were fed diets mixed with Cr2O3 as exogenous indigestible marker to determine apparent digestibility of nutrients. Using the following formula: digestibility (%) = {1 − [(Nf × Cd)/(Nd × Cf)]} × 100, where, Nf = nutrient concentration in faeces (% DM), Nd = nutrient concentration in diet (% DM), Cd = chromium concentration in diet (% DM) and Cf = chromium concentration in faeces (% DM).
Point 5: In Discussion some paragraphs alude other by-products, however did not explain how chemical components included also in Fermented tea residue do work. Avoid speculations.
Response 5: Thanks for your suggestion, the discussion has been deeply changed and added how the chemical components in fermented tea residues affect pigs (Line 283-291).
We tried our best to improve the manuscript and made some changes in the manuscript. These changes will not influence the content and framework of the paper.
We appreciate for Editors/Reviewers’ warm work earnestly, and hope that the new version of the Manuscript fully matches to the requests of the Reviewers and of the Journal.
Once again, thank you very much for your comments and suggestions.
We look forward to hearing from you and to responding to any further questions and comments you and/ or the Reviewers may have.
Sincerely yours
Xaiqing Ding
on behalf of the authors

Reviewer 2 Report
General Comments
The manuscript negotiates a topic that may have interest for the pig industry, in terms of finding alternative ingredients for pigs diets. The authors however do not explain the rationale of the study and the product with special focus for pigs. In addition there are several drawbacks in the setup of the study, especially regarding the diet formulations, that are not explained by the authors. These details are not discussed in the manuscript. The number if replicates is also low, a factor which is not also explained by the authors, also in regard to the limitations of the study. Extensive English language editing is also required.
Specific Comments
Line 17: avoid using words as adjectives that may show exaggeration “cause great waste of resources”
Line 23: correct plural “ingredients”
Lines 27-29: in which growth stage the pigs were (starting bw in kg)?
Lines 32-33: “tended to increase” but the level of significance at the end of the sentence is at significant level. Such mistakes should have been avoided
Line 35: “drip loss and shear force of the treatment groups were significantly reduced” in which treatment and compared to which one? Same comment for lines 35-37, is this effect for all the groups supplemented with the product?
Lines 35-36: “Crude protein (CP), ether extract (EE), calcium (Ca) and phosphorus (p) levels were higher” levels in which parameter and material investigated? Such information should have been defined
Line 41: “VH” define abbreviation
Line 45: how digestion performance was estimated?
Lines 62-82: throughout this part of the Introduction the general properties of this by-product is presented, with some references of studies in other species. However, the authors should have elaborated on the properties of the product in relation to the nutrient requirements and feeding of pigs, why such a material could be helpful, in which context, why is important to seek for products that may assist in improving properties of the pigs meat, etc. The Introduction requires major revision.
Line 108: replicate number is rather low and limited. Therefore the conclusions drawn from the present study may not be regarded as definite and surely this has to be addressed in the Abstract and discussion and conclusions part
Lines 110-114: could the authors elaborate on the concept of reformulation of the diets in the experiment? What is the CP content of the product applied? What was the CP content of the soybean that was replaced? The reformulation as presented in Table 1, results in diets with a higher CP content and lower energy. Is this procedure compatible with optimal feeding guidelines of fattening pigs?
Lines 291-312: the discussion regarding growth performance requires major revision, as the authors do not address and justify the reason why the effects were observed, especially in relation with the reformulated diets that were used in the experiment.
Lines 337-340: a higher amount of protein was available via the feed in the treated groups with the tea-product, and this should be discussed in relation to the result observed in the digestibility. Diets were not iso-nitrogenous, and probably not with the same fat and energy content.
Lines 401-408: conclusion is not well written, needs major revision, addressing comments above as well. Also limitation due to replicate numbers is not stated.
Author Response
January 11, 2020
To the Editorial Office of the journal Animals
Dear Professors :
Thank you the reviewers for their constructive comments concerning our manuscript “Effects of fermented tea residue on fattening performance, meat quality, digestive performance, serum antioxidant capacity, and intestinal morphology in fattening pigs” (ID: animals-672334). Those comments are very valuable and helpful for revising our paper and guiding our researches. We have studied those comments carefully and have made corrections which we hope meet with approval. Revised portions are marked colored in the paper. The following is a point-to-point response to the editors' comments and recommendations. The main corrections in the paper and the response to the reviewer’s comments are as following:
Response to Reviewer 2 Comments and Reviewer 2’s comments were highlighted by using the Green Colored.
The manuscript negotiates a topic that may have interest for the pig industry, in terms of finding alternative ingredients for pigs diets. The authors however do not explain the rationale of the study and the product with special focus for pigs. In addition there are several drawbacks in the setup of the study, especially regarding the diet formulations, that are not explained by the authors. These details are not discussed in the manuscript. The number if replicates is also low, a factor which is not also explained by the authors, also in regard to the limitations of the study. Extensive English language editing is also required.
Specific Comments
Point 1: Line 17: avoid using words as adjectives that may show exaggeration “cause great waste of resources”; Line 23: correct plural “ingredients”; “VH” define abbreviation
Response 1: Thank you for your comment. We have carefully checked our manuscript and have revised the mistakes and problems of language the whole paper according to your suggestion and including some unreasonable statements and expressions (Line 17-18, Line 24, Line 42). All changes to our manuscript were highlighted by using the Green Colored Text.
Point 2: Lines 27-29: in which growth stage the pigs were (starting bw in kg)?
Response 2: with an average body weight 70 ± 1.0 kg were randomly selected (Line 29).
Point 3: Lines 32-33: “tended to increase” but the level of significance at the end of the sentence is at significant level. Such mistakes should have been avoided
Response 3: Thank you for your suggestion. We are very sorry for our incorrect writing. We have revised unreasonable statements and expressions the whole paper according to your suggestion (Line 34).
Point 4: Line 35: “drip loss and shear force of the treatment groups were significantly reduced” in which treatment and compared to which one? Same comment for lines 35-37, is this effect for all the groups supplemented with the product?
Response 4: Thank you for your comment. We have changed the unreasonable expression in the text, the changes are as follows: Compared with the CG, dietary supplementation of FTR significantly increased the nutrient digestibility of crude protein (CP), ether extract (EE), calcium (Ca) and phosphorus (P), improved the lipase and trypsin activities, reduced drip loss and shear force of fattening pigs (p <0.05) (Line 36-40).
Point 5: Lines 35-36: “Crude protein (CP), ether extract (EE), calcium (Ca) and phosphorus (p) levels were higher” levels in which parameter and material investigated? Such information should have been defined;
how digestion performance was estimated?
Response 5: We mentioned in the last week of the feeding experiment (2.5. Digestibility Trial and Digestive Enzyme), samples of feed-based diets were first collected for the determination of nutritional content, and then the nutritional content of air-dried fecal samples was determined, finally, the digestive performance of pigs is mainly estimated by investigating DM, CP, EE, Ca, P, OM in diet and feces. Parameters and materials are basic diets and feces from fattening pigs. pigs were fed diets mixed with Cr2O3 as exogenous indigestible marker to determine the dry matter (DM) crude protein (CP), ether extract (EE), calcium (Ca) and phosphorus (P) in diets and feces. Using the following formula: digestibility (%) = {1 − [(Nf × Cd)/(Nd × Cf)]} × 100, where, Nf = nutrient concentration in faeces (% DM), Nd = nutrient concentration in diet (% DM), Cd = chromium concentration in diet (% DM) and Cf = chromium concentration in faeces (% DM).
Point 6: Lines 62-82: throughout this part of the Introduction the general properties of this by-product is presented, with some references of studies in other species. However, the authors should have elaborated on the properties of the product in relation to the nutrient requirements and feeding of pigs, why such a material could be helpful, in which context, why is important to seek for products that may assist in improving properties of the pigs meat, etc. The Introduction requires major revision.
Response 6: We have deeply changed the Introduction taking into consideration your suggestions, hoping that now you can appreciate it more (Line 64-81).
Point 7: Line 108: replicate number is rather low and limited. Therefore the conclusions drawn from the present study may not be regarded as definite and surely this has to be addressed in the Abstract and discussion and conclusions part
Response 7: Thank you very much for your constructive comments and suggestion. We are very sorry for our replicate number is low, due to the limitation of the feeding environment conditions, we only had four replicates per group. We have modified the discussions, and conclusions according to your suggestion (Line 273-290, Line 380-386). In addition, we have also consulted many references on the effects of fermented feed on fattening pigs and found many experimental designs similar to our current experimental study. For example, Lei (2011) studied that Aspergillus awamori-fermented mung bean seed coats enhance the antioxidant and immune responses of weaned pigs. A total of 96 cross-bred weaned pigs were evenly assigned to four diet. 6 weaned piglets in four replicates per diet. Gyo (2011) researched that Effects of fermented mushroom (Flammulina velutipes) by-product diets on growth performance and carcass traits in growing-fattening Berkshire pigs. 225 Berkshire pigs were assigned to 5 dietary treatments, the five dietary treatments were divided among groups of 15 pigs with three replications. Similar experimental designs have also appeared in reports by Ali (2019), Ajay (2005), etc. This may be related to feeding conditions, feeding costs, etc. In the future, we will further explore whether fermented tea residues will be similar to our experiments when applied to large-scale fattening pigs.
Point 8: Lines 110-114: could the authors elaborate on the concept of reformulation of the diets in the experiment? What is the CP content of the product applied? What was the CP content of the soybean that was replaced? The reformulation as presented in Table 1, results in diets with a higher CP content and lower energy. Is this procedure compatible with optimal feeding guidelines of fattening pigs?
Response 8: Our previous research has shown that the crude protein, reducing sugar, protease of tea residue is significantly improved after fermentation by microbial strains. According to laboratory tests, the nutritional indicators were 41.20% crude protein, 4.71% crude fat, 11.08% crude fiber, 11.04% crude ash, 4.37% calcium, and 0.56% total phosphorus. Also contains chemical ingredients such as polyphenols, catechins, proteins, amino acids, tea saponins, vitamins etc. (Ding et al.,2019. Pending papers). Prior to the start of this experiment, we conducted an exploratory experiment to explore the range of addition of fermented tea residues and the range of alternatives of corn and soybean meal. It was concluded that the range of fermented tea residue was 10% -20% for the best growth of fattening pigs, so we used 10%, 15%, and 20% of fermented tea residue to replace corn and soybean meal. The replacement range of corn and soybean meal is 5%, 7.5%, and 10% in order to reduce the crude protein gap in nutrition. In the future, we will continue to explore the optimal amount of fermented tea residue and the best substitute for corn meal, in order to obtain the maximum effect. We are very sorry for our negligence of the CP content of the soybean has not been determined. The experimental basic diet was prepared in accordance with nutrient requirements established by the National Research Council [NRC 2012]. Although the reformulation as presented in Table 1, results in diets with a higher CP content and lower energy, the procedure compatible with optimal feeding guidelines of fattening pigs.
Point 9: Lines 291-312: the discussion regarding growth performance requires major revision, as the authors do not address and justify the reason why the effects were observed, especially in relation with the reformulated diets that were used in the experiment.
Response 9: The growth performance has been completely re-written following your opinion and suggestions (Line 274-291). We hope the quality of the manuscript will be considered better than before.
Point 10: Lines 337-340: a higher amount of protein was available via the feed in the treated groups with the tea-product, and this should be discussed in relation to the result observed in the digestibility. Diets were not iso-nitrogenous, and probably not with the same fat and energy content.
Response 10: We have revised this part according to your suggestion (Line 315-317).
Point 11: conclusion is not well written, needs major revision, addressing comments above as well. Also limitation due to replicate numbers is not stated.
Response 11: Thank you for the suggestion. The conclusion has been re-written (Line 381-387).
We tried our best to improve the manuscript and made some changes in the manuscript. These changes will not influence the content and framework of the paper.
We appreciate for Editors/Reviewers’ warm work earnestly, and hope that the new version of the Manuscript fully matches to the requests of the Reviewers and of the Journal.
Once again, thank you very much for your comments and suggestions.
We look forward to hearing from you and to responding to any further questions and comments you and/ or the Reviewers may have.
Sincerely yours
Xaiqing Ding
on behalf of the authors

Reviewer 3 Report
First I want to write: good work, however I don't like the description. I recommend rewriting two parts: Material and methods (2.2; 2.3; 2.4) and Results. I attached the manuscript where I marked some words, phrases or sentences which I don't like or I think that authors did a mistake - consider it. Sometimes I added comments to marked text.
I don't like "finishing pigs" term, better is "fatteners"/by the way shorter/, if you are mad about it, use both alternatively.
Introduction is a good written part. I found there an interesting approach of the by-tea product use. If we recommend use it as a feed ingredient I suppose that in future practice ruminants are better group of animals because of fiber content. Obtained good results in pigs could be connected with fermentation process and authors should shortly mentioned it here.
Materials and methods
- where are you obtain tea residue to prepare /ferment/ experimental feed ingredient from?
- grammar: check time - it is used alternatively Simple Present and Simple Past, sometimes you "lost" full senetence; latin names should be in italics
- perhaps it will be better: Tab.1 Composition and nutrition value of experimental feed; and add the line above the first one and write: experimental groups above CK, T1, T2, T3 - full description is written ealier and you don't need to repeat it below the table. In nutritional value part can't you add fiber content?
Results
- in Materials and Methods you presented abbreviations so use only them
- consider changes of all tables titles, below them don't repeat information from Methods
Discussion
- correct order of cited literature and check if from 43 number /which doesn't exist!/ everything is ok
This part of sentence: "the supplementation of fermented tea residues can improve the intestinal environment, and thereby increase the apparent digestibility of feed" written in conclusions should be moved to Discussion and please add something more. It is important notice. It can't be in Conclusions because you didn't investigate microflora /or intesitnal environment/ - you can't do everything, it is not a reservation.
References
- correct some marked mistakes and literature ordering from 43 point
Once again it was good work, good luck.

Author Response
January 11, 2020
To the Editorial Office of the journal Animals
Dear Professors :
Thank you the reviewers for their constructive comments concerning our manuscript “Effects of fermented tea residue on fattening performance, meat quality, digestive performance, serum antioxidant capacity, and intestinal morphology in fattening pigs” (ID: animals-672334). Those comments are very valuable and helpful for revising our paper and guiding our researches. We have studied those comments carefully and have made corrections which we hope meet with approval. Revised portions are marked colored in the paper. The following is a point-to-point response to the editors' comments and recommendations. The main corrections in the paper and the response to the reviewer’s comments are as following:
Response to Reviewer 3 Comments and Reviewer 3’s comments were highlighted by using the Yellow Colored.
Point 1: First I want to write: good work, however I don't like the description. I recommend rewriting two parts: Material and methods (2.2; 2.3; 2.4) and Results. I attached the manuscript where I marked some words, phrases or sentences which I don't like or I think that authors did a mistake - consider it. Sometimes I added comments to marked text.
Response 1: We have modified Material and methods (2.2; 2.3; 2.4) and rewritten Results (Line 206-209, Line 216-220, Line 226-230, Line 236-239, Line 245-249, Line 256-264). In addition, we have also revised the manuscript where you marked some words, phrases or sentences according to your suggestion. Any change to our manuscript following your suggestion was highlighted by using the Yellow Colored. Text.
Point 2: I don't like "finishing pigs" term, better is "fatteners"/by the way shorter/, if you are mad about it, use both alternatively.
Response 2: We have changed all “finishing pigs” in the manuscript to “fattening pigs” according to your suggestion.
Point 3: Introduction is a good written part. I found there an interesting approach of the by-tea product use. If we recommend use it as a feed ingredient I suppose that in future practice ruminants are better group of animals because of fiber content. Obtained good results in pigs could be connected with fermentation process and authors should shortly mentioned it here.
Response 3: The introduction has been modified (Line 81-82) according to the suggestion of all of you, hoping that now you can appreciate it more. Since the technological aspect represent a novelty of the manuscript, we prefer to maintain it in the introduction.
Point 4: where are you obtain tea residue to prepare /ferment/ experimental feed ingredient from?
Response 4: Thank you for your comment. We have already mentioned about obtaining tea residue for preparation, fermentation, and supplementing the experimental feed ingredients in the text (2.1. Fermented Tea Residue Sample), (Line 95-107). Our previous research has shown that the crude protein, reducing sugar, protease of tea residue is significantly improved after fermentation by Bacillus subtilis, Aspergillus niger and Saccharomyces cerevisiae. According to laboratory tests, the nutritional indicators were 41.20% crude protein, 4.71% crude fat, 11.08% crude fiber, 11.04% crude ash, 4.37% calcium, and 0.56% total phosphorus. Also contains chemical ingredients such as polyphenols, catechins, proteins, amino acids, tea saponins, vitamins etc. (Ding et al.,2019. Pending papers).
Point 5: grammar: check time - it is used alternatively Simple Present and Simple Past, sometimes you "lost" full senetence; latin names should be in italics
Response 5: Thank you for your comment. We have carefully revised the grammar and some detail errors in our manuscript and marked it with different colors according to all the referee suggestions.
Point 6: perhaps it will be better: Tab.1 Composition and nutrition value of experimental feed; and add the line above the first one and write: experimental groups above CK, T1, T2, T3 - full description is written ealier and you don't need to repeat it below the table. In nutritional value part can't you add fiber content?
Response 6: We have changed Tab.1 to “Composition and nutrition value of experimental feed” (Line 124). And have added the line above the first one and write: experimental groups above CK, T1, T2, T3 according to your suggestion. In addition, in this test, the crude fiber content of tea residue after fermentation is 11.08%, about why the fiber content is not added to the nutritional value part, we consulted a large number of references when designing this experiment, and designed this experiment based on many previous studies. (For example, Chang et al. did not add crude fiber content to the nutritional value part when studying the effects of Aspergillus niger fermented rapeseed meal on nutrient digestibility, growth performance and serum parameters in growing pigs. Gyo et al. explored the effects of fermented mushroom by-product diets on growth performance and carcass traits in growing-fattening Berkshire pigs, the crude fiber was also not added to the nutritional value part, and the same experimental design was also reported by Lee and Olstorpe), so we also adopted an experimental design similar to them. Crude fiber content is not added to the nutritional value part.
Point 7: Materials and Methods you presented abbreviations so use only them
Response 7: Thank you for your suggestion. We have modified them in the text.
Point 8: consider changes of all tables titles, below them don't repeat information from Methods.
Response 8: We have changed the titles of all the tables and below them removed repeat information from Methods according to your suggestion.
Point 9: correct order of cited literature and check if from 43 number /which doesn't exist!/ everything is ok
Response 9: Thank you for your suggestion. We have changed the order of citations from 43. And rechecked the citations of the literature.
Point 10: This part of sentence: "the supplementation of fermented tea residues can improve the intestinal environment, and thereby increase the apparent digestibility of feed" written in conclusions should be moved to Discussion and please add something more. It is important notice. It can't be in Conclusions because you didn't investigate microflora /or intesitnal environment/ - you can't do everything, it is not a reservation.
Response 10: Thank you for your suggestion. We are very sorry for our incorrect writing. We have rewritten the conclusions, moved some of the previous conclusions into the discussion, and add something according to your suggestion (Line 376-379, Line 381-387).
Point 11: correct some marked mistakes and literature ordering from 43 point
Response 11: Thank you very much for your constructive comments and your valuable time,We have double checked the text and corrected some marked mistakes and literature ordering from 43 point.
We tried our best to improve the manuscript and made some changes in the manuscript. These changes will not influence the content and framework of the paper.
We appreciate for Editors/Reviewers’ warm work earnestly, and hope that the new version of the Manuscript fully matches to the requests of the Reviewers and of the Journal.
Once again, thank you very much for your comments and suggestions.
We look forward to hearing from you and to responding to any further questions and comments you and/ or the Reviewers may have.
Sincerely yours
Xiaoqing Ding
on behalf of the authors

Round 2
Reviewer 2 Report
The revised version has been improved compared to the initial submitted one.
The authors have complied to most points addressed and corrections have been made.
Author Response
January 15, 2020
To the Editorial Office of the journal Animals
Dear Professors :
Thanks very much for your kind work and consideration concerning our manuscript “Effects of fermented tea residue on fattening performance, meat quality, digestive performance, serum antioxidant capacity, and intestinal morphology in fattening pigs” (ID: animals-672334). We have carefully checked the language and spelling errors in the manuscript to further improve our manuscript, which we hope meet with approval. On behalf of my co-authors, we would like to express our great appreciation to editor and reviewers.
Thank you and best regards.
Yours sincerely,
Xiaoqing Ding
Reviewer 3 Report
I found improved the manuscript but still there is some language erros, so once again you have to read the text and correct it.
Once again I attached the manuscript where I marked some words or phrases (not all - looking for them it is your work) which should be corrected.
Once again look into tables: if in Tab 1 you add line with "experimental groups" be consistent later (in other tables) and remove additional explanations.
To pointe the mistakes it is my duty but still I think that it is a good work,
you have to improve presentation of results - language

Author Response
January 15, 2020
To the Editorial Office of the journal Animals
Dear Professors :
Thanks very much for your kind work and consideration concerning our manuscript “Effects of fermented tea residue on fattening performance, meat quality, digestive performance, serum antioxidant capacity, and intestinal morphology in fattening pigs” (ID: animals-672334). Those comments are very valuable and helpful for revising our paper and guiding our researches. We have studied those comments carefully and have made corrections which we hope meet with approval. Revised portions are marked Yellow Colored in the paper. The following is a point-to-point response to reviewer 3’s comments.
Point 1: I found improved the manuscript but still there is some language erros, so once again you have to read the text and correct it.
Response 1: Thank you very much for your constructive comments and suggestion. We have carefully checked our manuscript and have revised the mistakes and problems of language the whole paper according to your suggestion and including some unreasonable statements and expressions.
Point 2: Once again I attached the manuscript where I marked some words or phrases (not all - looking for them it is your work) which should be corrected.
Response 2: Thanks again for your hard work. we have revised the manuscript where you marked some words, phrases or sentences according to your suggestion. In addition, we also rechecked the unreasonable expression in the manuscript and corrected it
Point 3: Once again look into tables: if in Tab 1 you add line with "experimental groups" be consistent later (in other tables) and remove additional explanations.
Response 3: We have added “experimental groups” to all tables and removed additional explanations according to your suggestion.
Point 4: To pointe the mistakes it is my duty but still I think that it is a good work, you have to improve presentation of results - language
Response 4:We tried our best to improve presentation of results. These changes will not influence the content and framework of the paper. We appreciate for Reviewers’ warm work earnestly, and hope that the new version of the Manuscript fully matches to the requests of the Reviewers and of the Journal.
On behalf of my co-authors, we would like to express our great appreciation to editor and reviewers.
Thank you and best regards.
Yours sincerely,
Xiaoqing Ding